# On the potential of using smartphone sensors for wildfire hazard estimation through Citizen Science

Hofit Shachaf and Colin Price
Porter School of the Environment and Earth Sciences, Tel Aviv University, Tel Aviv, 69978, Israel

Dorita Rostkier-Edelstein
The Fredy and Nadine Herrmann Institute of Earth Sciences, The Hebrew University of Jerusalem

Cliff Mass
Department of Atmospheric Sciences, University of Washington, Seattle, USA

*Correspondence to*: Colin Price (colin@tauex.tau.ac.il)

**Abstract**

**Weather conditions that can enhance wildfire potential are a problem faced by many countries around the world. Wildfires can have major economic impacts as well as prolonged effects on populations and ecosystems. Distributing information on fire hazards to the public and first responders in real-time is crucial for fire risk management and risk reduction. Although most fires today are caused by people, weather conditions determine if and how fast the fire spreads. In particular, research has shown that atmospheric vapor pressure deficit (VPD) is a key parameter predicting the dryness of vegetation and the available fuel for fires. VPD is determined from the environmental air temperature and relative humidity, both of which are readily obtained from smartphones carried by the public. In this study we use smartphone data from the OpenSignal company, collected during almost 4 years and from more than 40,000 users per day, to estimate VPD values. We have found that smartphone data can provide useful information about fire risk and danger. Here we present two case studies from wildfires in Israel and Portugal in which VPD is calculated using calibrated temperature and relative humidity measurements from smartphones. Given the rapid growth in the number of smartphones around the globe, we propose applying smartphone data for meteorological research and fire-weather applications. Possible users of these results could be wildfire researchers; public policy specialists in wildfire, climate and disaster management; engineers working with big data; low-income countries; and citizen science advocates.**

## 1. Introduction

In the past years there has been a dramatic rise in wildfires across the globe. Between January 2017 and August 2017 there were around 40,000 wildfires in the US that burned approximately 2.3 million hectares. In California and Portugal, 2017 was the worst wildfire season on record, with major destruction of homes and natural vegetation. During November 2019 Australia restricted its outdoor water consumption to maintain sufficient water supply to help firefighters, and in 2019 California wildfire damage was estimated at more than $80 billion. In 2023 there were record fires across Canada, with the resulting air pollution and smoke spreading across the northeast U.S. It is clear

that improved monitoring and forecast models will foster better early wildfire warnings that will allow people, cities and countries to be better prepared, with implications for reducing the loss of lives and damage to property and infrastructure.


Monitoring such extreme weather hazards has typically used traditional weather stations and sensors for in situ measurements of the environment. In this study we propose that micro-sensors in smartphones carried by the public may provide additional and highly complimentary data. The development of smartphones during the last two decades, and the reduction in their cost, has led to more than 6.4 billion smartphones in use worldwide today (out of

almost 8 billion estimated people). Today, smartphones are often more accessible to the population in some countries than electricity or running water, and the global distribution of smartphones, together with improved internal sensors, is only expected to increase during the coming decades.

In recent years, several research groups have used smartphone data in scientific research. In the Netherlands,

smartphones have been used to study air pollution (Snik et al., 2014), while in the United States smartphones have been used to study atmospheric pressure variability and the potential for improved numerical weather prediction (Mass and Madaus, 2014; McNicholas and Mass, 2021). Droste et al. (2017) and Hintz et al. (2019) showed that smartphone pressure sensors are a reliable source of atmospheric data, with biases of the order of 1 mb. In the UK, a study demonstrated that smartphones can be used to map temperature changes and anomalies (Overeem et al., 2013),

and in Israel (Price et al., 2018) smartphone data were used to study semi-diurnal tides in the atmosphere.

The conditions that can affect the ignition and propagation of wildfires have been studied for more than a century, and can be influenced by large-scale climate phenomena such as the El Nino Southern Oscillation (ENSO) and the Indian Ocean Dipole (Goldammer and Price, 1998; Bovalo et al., 2012). However, on a daily basis, fire weather

models use surface weather conditions (temperature, humidity, wind speed, and precipitation) on hourly to daily scales to generate fire weather indices (Baumgartner, 1967; McArthur, 1967; Keetch and Byram, 1968; Kase, 1969; Fosberg, 1978; Anderson, 1982; Chandler et al., 1983; Van Wagner and Forest, 1987; Sharples et al., 2009; Di Giuseppe et al., 2020). Some studies have combined large scale climate indices and local meteorological parameters (Shen et al., 2019) to estimate fire risk and danger on longer time scales.


One of the main criteria for estimating fire danger and behavior is the moisture content of vegetation (Keetch and Byram, 1968; Schroeder and Buck, 1970; Anderson, 1982; Sharples et al., 2009; Shen et al., 2019). Indices like the Fire Potential Index (FPI), Thousand-Hour Fuel (TH), Dead Fuel Moisture (DFM) and others, are used to predict fire risk (Hernandez-Leal et al., 2006; Escuin et al., 2008). Fire indices vary in their complexity; some are very simple

like the Angstrom index (Chandler et al., 1983), which uses only temperatures (T) and relative humidity (RH), while others are more complex, applying additional meteorological parameters, along with soil properties and the biological life cycle of the plants like M68 (Kase, 1969). However, the majority of indices are based on the two key parameters, **temperature** and **relative humidity** (Table 1). While daily and hourly meteorological data can be

obtained using traditional measuring sensors, smartphones potentially offer an additional source of reliable data
(Overeem et al., 2013; Mass and Maudus, 2014; Fujinami, 2016, Hintz et al., 2019). Furthermore, smartphones can
theoretically supply high spatial resolution of the observed parameters. For example, Figure 1 shows the spatial
coverage of smartphone readings on one day (4 June 2014) in Israel compared to the official weather station
distribution from the Israel Meteorological Service IMS stations.

**Table 1. Summary of input data used in the most common fire danger indices. The abbreviations for the meteorological parameters are T- temperature, RH - relative humidity, P - precipitation, U – winds speed.**

| Fire indices | Acronym | Meteorological data | Other parameters | References |
|---|---|---|---|---|
| **Angstrom Index** | **Angstrom** | *T, RH* | | [12] |
| **Boumgartner index** | **Boumgartner** | *T, Tmin, Tmax, P, U* | *Elevation ,latitude* | [13] |
| **Fine fuel moisture code** | **FFMC** | *T, RH, P, U* | | [10] |
| **Duff moisture code** | **DMC** | *T, RH, P* | | [10] |
| **Drought code** | **DC** | *T , P* | | [10] |
| **Initial spread index** | **ISI** | *T, RH, P, U* | | [10] |
| **Bulidup index** | **BUI** | *T, RH, P* | | [10] |
| **Fire weather index** | **FWI** | *T, RH, P, U* | | [10] |
| **Fosberg fire weather index** | **FFWI** | *T, RH, U* | | [14] |
| **Keetch- Byram drought Index** | **KBDIsi** | *T, P, P(annual)* | | [15] |
| **McArthour Mark 5 forest fire danger index** | **FFDI** | *T, RH, P, P(annual), U* | | [16] |
| **Sharples fuel moisture index** | **FMI** | *T, RH* | | [18] |
| **M68 index** | **M68** | *T, RH, P, U* | *snow, phenology* | [17] |

The vapor pressure deficit (VPD) can be calculated from temperature and the relative humidity.  Jain et al. (2022)
have used VPD to show that a decrease in RH and an increase in T were primarily responsible for increases in
extreme fire weather conditions globally. Other fire-related VPD studies done in different parts of the world, like the
study done by Park et al. (2014) in the southwestern United States (SWUS), found that in spring and early summer
(March- July) 1961-2014 the average VPD was ~15 hPa. When they compared it with the same months of 2011,
which is considered as a year of extreme drought with record breaking wild fires, they found VPD anomalies of +3
hPa.  Their finding showed that even though it was not exceptionally hot in the southwest US, it was exceptionally
dry, showing that annual burnt area is closely related to spring-summer potential evapotranspiration and VPD
anomalies.  Thus, monitoring T and RH, and hence VPD, from smartphone sensors have the potential for providing
useful information about VPD at high spatial resolution and high temporal resolution even in remote areas with few
official weather stations. Crowd-sourcing of smartphone data may therefore provide a new tool for analyzing the
risk of fires in real time.

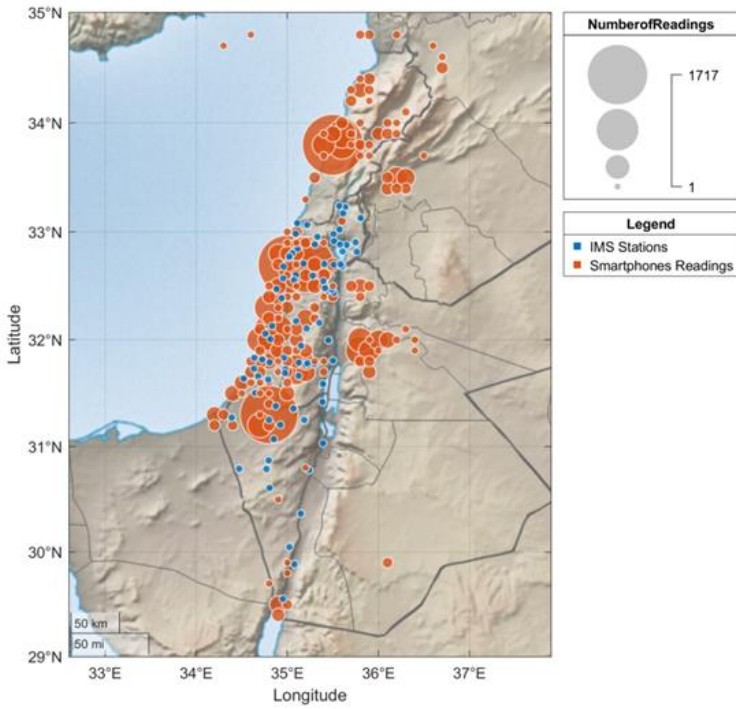


**Figure 1. Smartphones observation during one day (4 June 2014) with each circle representing the number of readings in a 0.1˚ x 0.1˚ grid box. The IMS station coverage (official reference stations) is shown by uniform size blue dots. The number of data samples from IMS is 86,400 samples on this day. The total number of data samples from smartphones was 15,770.**

## 2. Methodology

### 2.1 Calibration Methods

Most smartphones today have a set of sophisticated micro-sensors that measure several local environmental parameters. The most common sensors in smartphones measure atmospheric pressure, magnetic field, light, temperature, relative humidity, GPS location, sound, and even gravity and acceleration in three directions. This

paper focuses on two parameters measured by smartphones: T and RH. RH sensors are usually capacitive sensors that measure RH by placing a thin strip of metal oxide between two electrodes. The metal oxide's electrical capacity changes with the atmosphere's RH (Yoo et al., 2010). The internal thermometer of smartphones do not directly measure ambient air temperature, and are impacted by heat sources within the smartphone. Thus, a major challenge is to estimate environmental air temperatures from the smartphone temperatures measured inside the unit.


Figure 2 shows our control experiment done in Israel comparing a stationary smartphone (Samsung Galaxy S4) and an adjacent weather station (Davis Vantage VUE) for both T and RH. The smartphone was placed in a fixed location next to the Davis system (less than a meter distance) on a table in a shaded room with open windows. The

temperature sensor used in the Samsung Galaxy S4 model is SHTC1 version 1 Sensirion
(https://www.sensirion.com/products/catalog/?category=Humidity) with a resolution of 0.01°C, an accuracy of 0.3°C
and a range between -40° to 125°C (Cabrera et al., 2021). The Davis temperature sensor has 0.1°C resolution with a
nominal accuracy of 0.5°C and ranges between -40° to 65°C. The same sensor (SHTCI1) is used in Samsung Galaxy
S4 for RH, with a resolution of 0.01%, accuracy between 3%-5% that depends both on temperature of the
environment and the humidity, and a range between 0% - 100%. The Davis humidity sensor has a resolution of 1%,
accuracy of 2% and range between 0% - 100%.

The data acquired from both sensors were split into training/learning data and testing data (Figure 2). Using simple
linear regression we calibrated the smartphone data using the Davis T and RH as ground truth for two periods (2-6
October and 7-10 October).    As can be seen in Figure 2, after calibrating with a simple linear regression, the
correlation between the calibrated T and RH data and the Davis data is very high for both parameters
($R^2$ $of$ 0.86 $and$ 0.975).  This experiment was repeated several times with several different devices (Galaxy S4)
and locations, and all show the same results, consistent with Price et al. (2018).

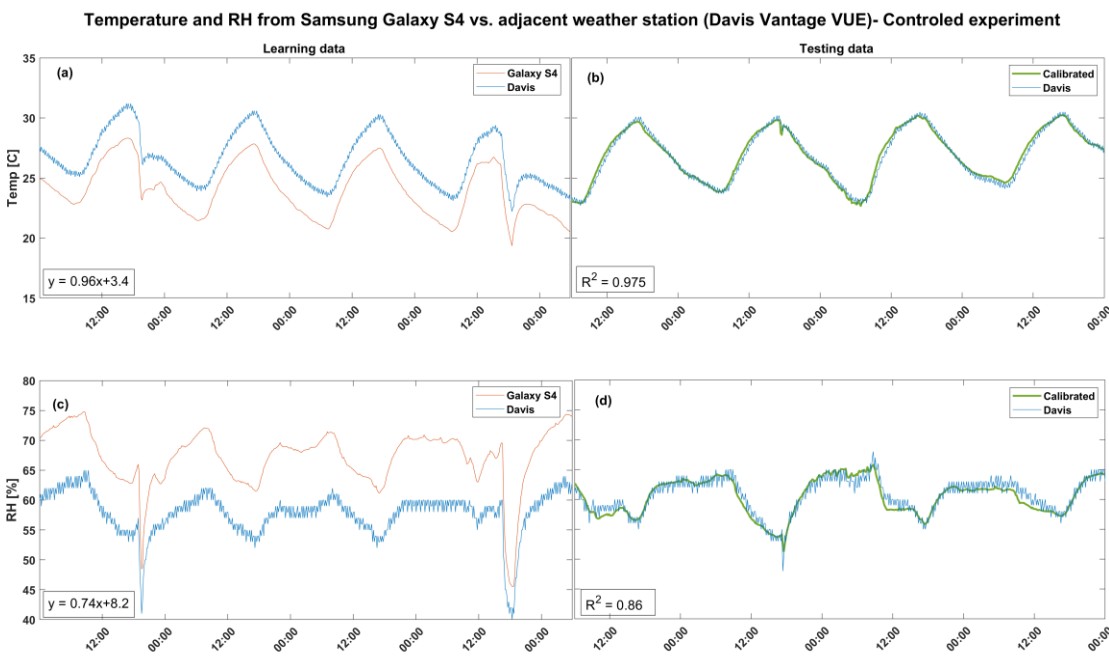

**Figure 2. Control experiment with a stationary smartphone (Samsung Galaxy S4) and an adjacent weather station (Davis Vantage VUE), conducted in Israel. (a) & (c) Raw smartphones (red) and weather station (blue) T & RH values, respectively, during 4 days (2-6 October). (b) & (d) Smartphone calibrated data (green) and weather station (blue) T & RH values respectively for a different 4 days (7-10 October).**


Given this good estimation for the changes in T and RH using four different Samsung Galaxy S4 smartphone
sensors, the next step was to determine how well crowd-sourced smartphone data (that are non-stationary and are
being used in different and varied environments) agree with measurements from official meteorological stations.

For the crowd-sourced data from smartphones we used data from the **WeatherSignal App** (operated by Open Signal

https://www.opensignal.com/).  Open Signal provided almost 4 years of smartphone data (2013-2016) originating from their WeatherSignal App and collected from more than 40,000 smartphone daily users (58,000 in 2014 and 40,000 in 2015) around the globe. Raw smartphone data were supplied without any phone or user ID, to protect user privacy.  We note that the WeatherSignal App does not operate on IOS (iPhone) and is available only for Android smartphones.

Although this appears to be a relatively large number of global daily measurements, the measurements for specific local locations can be highly variable and limited, and in this study we were limited to the coverage of WeatherSignal users.  Using smarphone data we calculated VPD on a 1 by 1 degree grid  using daily mean T and RH from smartphone data.  Since Overeem et al. (2013) already investigated crowd-sourced temperature data from the same WeatherSignal data, here we focus primarily on the quality of the RH data from WeatherSignal, although we also show our new analysis of temperature data from these same smartphones.

Due to the extreme fire season in the Iberian Peninsula during 2013, in Figure 3 we present a comparison between the RH data from two official meteorological stations in the south of Spain during 2013 (European Climate Assessment & Dataset - ECA&D) and the crowd-sourced RH data from smartphones in the same region. The RH data from the two weather stations in the south of Spain (latitude: 36.75 N, longitude: 6.0625 W and latitude: 36.5 N, longitude: 6.2625 W) were compared with all smartphone data collected in the same area (within latitudes 36- 37 N, longitudes 7 – 6 W). The daily RH in this region was determined using ~230 smartphone data points per day. After a 3-month period of training and calibration between the official stations and the smartphone data, Figure 3 shows the calibrated relative humidity data for June to December 2013. The blue curve indicates the RH daily mean from the two official weather stations, while the green curve indicates the RH daily mean from smartphones after calibration. The calibration was done also using a simple linear regression model.  The correlation $R^2$ between smartphone RH and official observations was greater than 0.7, implying that the smartphone data can explain more than 70% of the daily variability of the RH measured by the two meteorological stations in southern Spain.  The smartphone data averaged in space and time successfully duplicates the daily fluctuations in RH for this region.

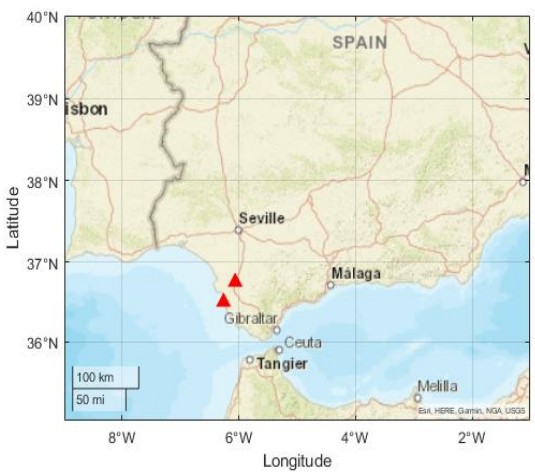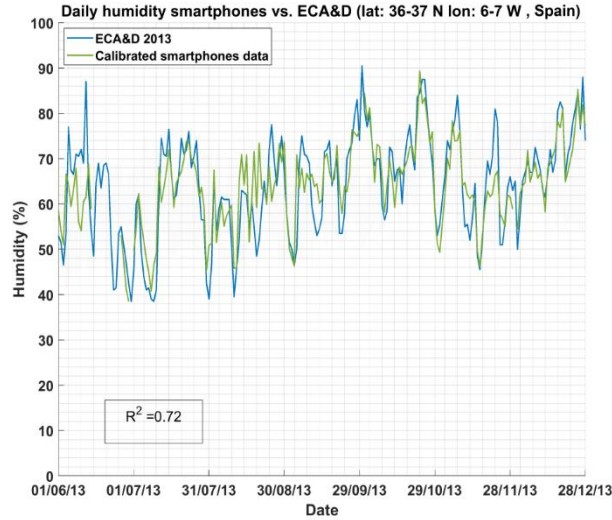

**Figure 3. RH data from two weather stations in Spain extracted from ECA&D marked with red triangles (left panel), with the exact location at 36.75 N, 6.0625 W and 36.5 N, 6.2625 W. The RH data collected from smartphones was collected within latitudes 36- 37 N and longitudes 7 – 6 W (over land). Smartphones were trained using data from 3 previous months in the datasets. The calibrated smartphone (green) and meteorological stations (blue) mean RH data are shown following the training period during June to December 2013 ($R^2$=0.72).**

A similar comparison for smartphone T and RH was performed using the European Centre for Medium-Range Weather Forecast (ECMWF) ERA5 reanalysis (Hersbach et al., 2020), and data from the Israeli Meteorological Services (IMS) as ground truth in Israel. Figure 4 and 5 show training/learning data (left panels) and testing data (right panels) for T and RH in Israel. Part of the temperature data sets (learning data) were used to establish the linear relation between the ERA5 and the smartphone data sets (Jun 2013 – Dec 2015), and then were applied to 2016 using the calibrated data (Figure 4). The same calibration process was done for RH data from Israel. Figure 5 shows the training/learning RH data and calibration equation for July and August 2016, which was then applied to September to Dec 2016 (testing data). In general RH data are noisier data compared with temperature data, with less regular diurnal and seasonal trends. Hence, we generally used shorter training data sets (months instead of years) for calibrating the RH data. The calibrations using both data sets (ERA5 and IMS) result in strong correlations between the calibrated data and ground-truths both for T and RH ($R^2 > 0.83$).

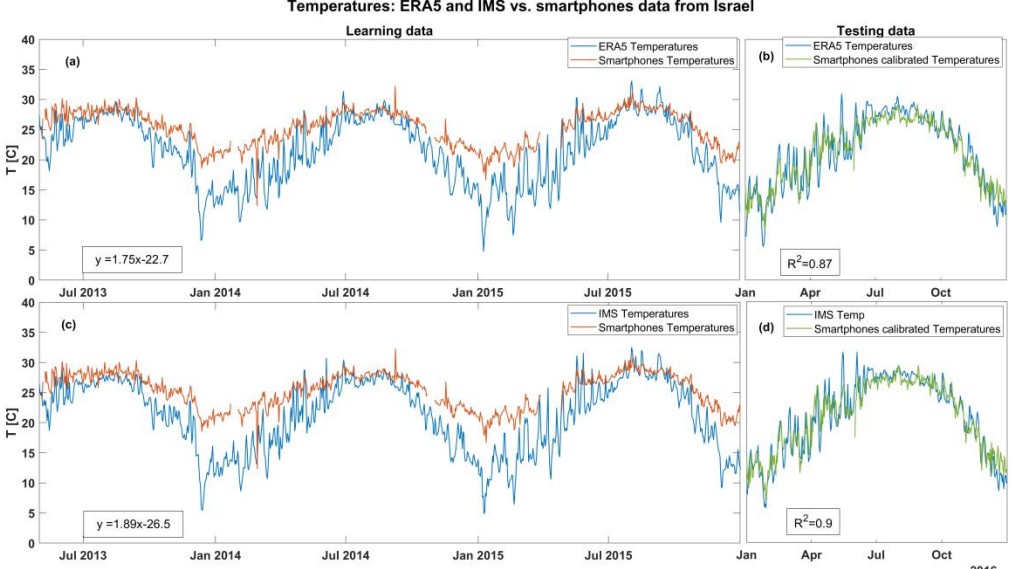

**Figure 4. Daily temperatures for Israel from ERA5, IMS and smartphones. (a) & (c) Learning data: linear regression equations using the two data sets of 2.8 years (blue- ERA/IMS, red- smartphones). (b) & (d) Calibrated smartphone data in green and ERA5 and IMS data in blue, for the year of 2016 ($R^2$= 0.87-0.9). The regression curve uses temperature in the units of Celsius.**

**Figure 5. Daily RH for Israel from ERA5, IMS and smartphones. (a) & (c) Learning data: linear regression equations were found using these two data sets during 2 months (blue- ERA/IMS, red- smartphones). (b) & (d) Calibrated smartphone data in green and ERA5 and IMS data in blue, for September-December 2016 ($R^2$= 0.83-0.97).**

A further comparison was done with T and RH data collected from ERA5 and smartphones in Portugal. The results are shown in Figure 6 and 7. Figure 6 shows the temperature learning data in the left panel (Jun 2013 – Jan 2015) and the testing data in the right panel, showing a high correlation coefficient ($R^2 > 0.8$). Figure 7 shows the RH

learning data in the left panel (Jun – Dec 2013) and the calibrated testing data in the right panel, showing a rather moderate correlation ($R^2 = 0.55$). As mentioned above, we find that the calibrated T data is generally in better agreement with the ERA5 data than the calibrated RH data. This could be due to fewer smartphones having RH sensors than temperature sensors, and hence there is a larger sensitivity to RH outliers, and/or the impacts of rapid changes of RH over relatively short distances between users (unlike temperature). In conclusion, we find that the RH crowd-sourced data are less reliable than the T crowd-sourced data, even though the temperature data are significantly impacted by the battery temperature in the phones.

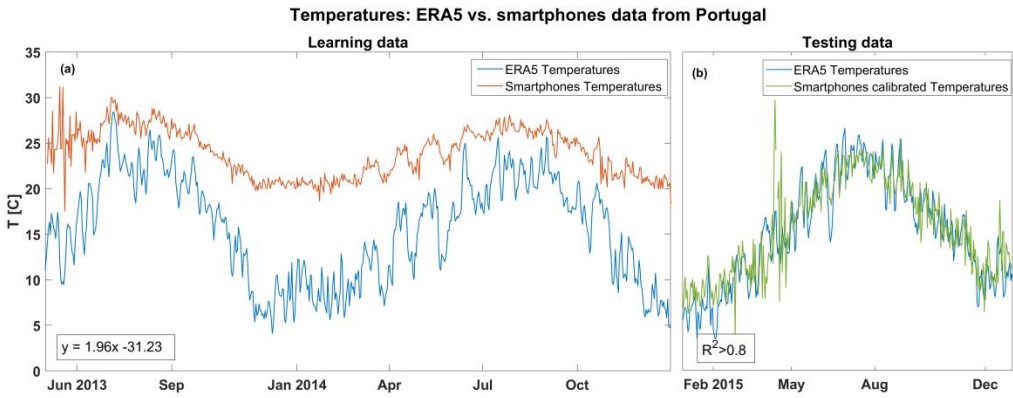

**Figure 6. Daily temperatures for Portugal from ERA5 and smartphones. (a) Learning data: linear regression equations were found using these two data sets over 1.6 years (blue- ERA, red- smartphones). (b) Calibrated smartphone data in green for the year of 2015 ($R^2 > 0.8$). The peak temperatures in April 2015 were associated with the extreme heat in the Iberian peninsula during April 2015 with temperatures reaching nearly 40C in some locations.**

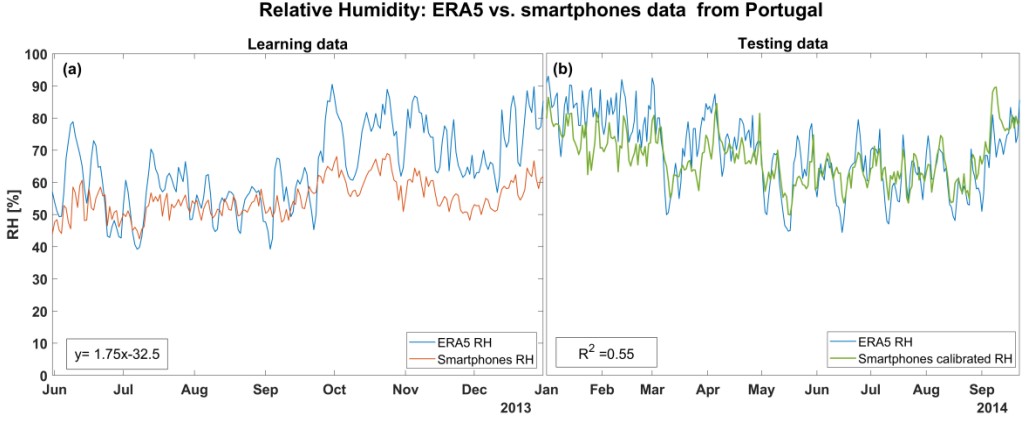

**Figure 7. Daily RH for Portugal from ERA5 and smartphones. (a) Learning data: linear regression equations were found using these two data sets over a 7-month period (blue- ERA, red- smartphones). (b) Calibrated smartphone RH data in green for the year of 2014 ($R^2 = 0.55$).**

From the temperature data (Figures 4 and 6) the annual variations in smartphone T values are apparent (lower T in winter time and higher T in summer) but compared with data from ECMWF and IMS data the winter temperatures are always too warm. This can be explained by smartphone T being affected by internal heat, and moreover, most smartphone temperature measurements in winter are taken indoors, as opposed to the official weather stations. In

addition, the ECMWF model has its own obvious biases in wintertime, often reporting warm biases too. So the
ERA5 reanalysis cannot always be regarded as "ground truth". Nevertheless, using the ECMWF and IMS data to
calibrate T improves dramatically the accuracy of the smartphone data. All further analyses were done with daily
calibrated smartphone data (T and RH).

## **2.2 Vapor Pressure Deficit (VPD)**

Vapor pressure deficit (VPD) is the difference between the water vapor content of the air and its saturation value
(Equation 1(a)). Unlike T and RH each by itself, VPD can reflect more accurately the ability of the atmosphere to
extract moisture from the land surface and fuels, and estimate the potential of fuel for fires. While RH is defined as
the ratio between the actual vapor pressure $e_a$ and the saturation vapor content of the air at a certain temperature, $e_s$
($T_a$), in percentage (Equation 1(b)), it is not an absolute measure like VPD (hPa). In addition, VPD shows an almost
linear relationship with the rate of evapotranspiration.

$$(a) \ \boldsymbol{VPD} = e_s(T_a) - e_a \quad (b) \ \ \boldsymbol{RH} = 100 \times \frac{e_a}{e_s(T_a)} \qquad \text{(1a and 1b)}$$

VPD can be calculated by using RH and $e_s(T_a)$, as can be seen in Equation 2:

$$VPD = e_s(T_a)(1 - \frac{RH}{100}) \qquad \text{(2)}$$

$$e_s(T_a) = 0.61094 \cdot exp(\frac{17.625T}{T+243.04}) \qquad \text{(3)}$$

The August-Roche-Magnuse approximation (Equation 3) presents an empirical relationship for $e_s$ that implies VPD
varies exponentially as a function of T and RH because $e_s(T_a)$ depends on the Clausius Clapeyron equation. In other
words, the same water vapor pressure at different temperatures results in very different RH values. Despite its low
popularity VPD has been investigated in several papers and has shown a high correlation with burned area for forest
fires in the U.S. (Park Williams et al., 2014; Seager et al., 2015; Sedano and Randerson, 2014; Brown et al., 2023;
Rao et al. 2023).

In our analysis we calculated VPD both temporally and spatially using the calibrated smartphone data (T and RH).
Daily VPD values were calculated for the entire time period as well as during large fire events. The background
climatology of VPD was calculated using ERA5 T and RH data from 2000 to 2010 (10 years). These years are
independent of our data (2013-2016) and represent a climatological background for comparison with the smartphone
data.

For the spatial analysis, a spatial anomaly index was created, using a climatology of VPD in 3 non-fire years from
the smartphone data, averging the daily VPD at a spatial resolution of 1˚ x 1˚ degree. When analysing the wildfire
case studies, we subtracted the daily mean VPD of a specific month from the 3–year non-fire *monthly mean* to

calculate the ΔVPD (VPD anomaly). An index of zero means that the daily VPD is the same as the monthly climatology, and a negative or positive index indicates that VPD is lower or higher than usual for this day. An anomaly larger than two standard deviations from the climatological VPD represents an statistically significant anomaly at the 95% level, while a three standard deviation anomaly represents a statistical significance greater than 99.7%. High positive values of VPD imply enhanced drying of vegetation, and enhanced fire risk.

## 3. Results


Below, we will present the analysis of VPD during wildfire events in Israel and Portugal between 2013 and 2016 using smartphone data.

### 3.1 Israel

From 18-29 November 2016, Israel was influenced by two different pressure systems causing dry surface winds
from the northeast. There were more than 1770 fires, 40 of which were considered mega fires (burning more than 4,000 hectares), houses and properties were destroyed, and around 300 people were injured. The total damage was estimated at $150 million (KKL JNF, 2020). Data from the IMS show that the overall mean values of the relative humidity during that period (coming from 80 stations scattered around Israel) were below 20% and at the peak was as low as 10%.


Figure 8 shows the temporal and spatial analysis on regional maps showing the absolute VPD (Figure 8) and the VPD anomaly (Figure 9) calculated from smartphones for individual days in November 2016. The maps start on 16 November, with a 3-day interval between each map. From 22-27 November widespread fires occurred, with the VPD evolving during these two weeks. As mentioned before, ΔVPD is calculated using the monthly mean of the
calibrated VPD for a specific month (here November 2013, 2014 and 2015) and subtracting Nov 2016 **daily means** from the Nov 3–year **monthly mean**. Both Figures 8 and 9 show extreme anomalies (> 8 hPa) in VPD during the days of wildfire (22-27 Nov).

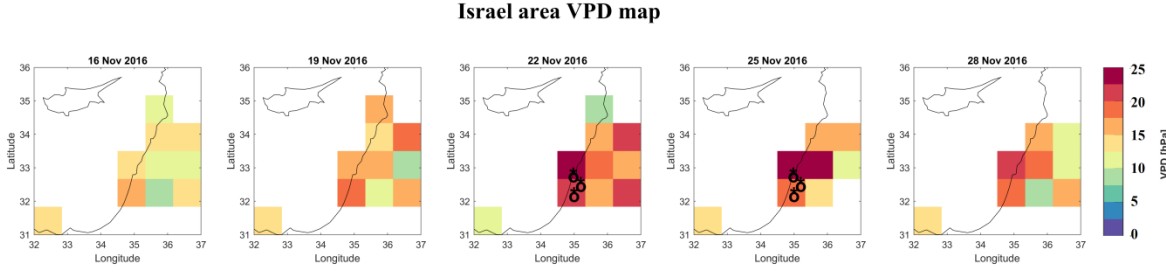

**Figure 8. Absolute VPD calculated from smartphones for days in November 2016, starting with 16 Nov and with a 3-day
interval between each map. Extreme wildfires occurred from 22-27 November (locations of fires are marked with black symbols).**

**Israel area ΔVPD map**

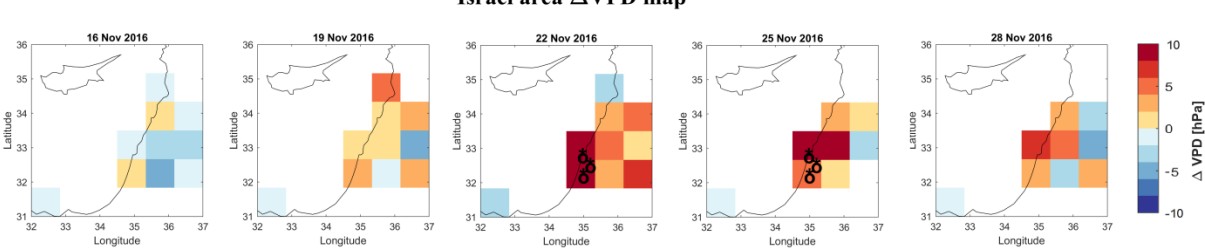

**Figure 9. ΔVPD calculated from smartphones for days in November 2016, starting with 16 Nov and with a 3-day interval between each map. Extreme wildfires occurred from 22-27 November (locations of fires are marked with black symbols).**


Figure 10 shows July to December 2016 daily VPD calculated from smartphones (red line) across Israel, compared with the 2000-2010 climatology of daily mean VPD for the same area from ERA5 (blue dash line), together with one standard deviation (light blue lines). A significant increase and anomaly in VPD is detected a day before the fires start (21 Nov 2016) and during the fires, with an increase that starts at 1σ going up to 8σ. Hence, this anomaly

in VPD in November, detected by the smartphones, is statistically significant at the 99.9999% level. There was also another large VPD positive anomaly at the end of August 2016, implying high danger for wildfires. However, no significant fires occurred.

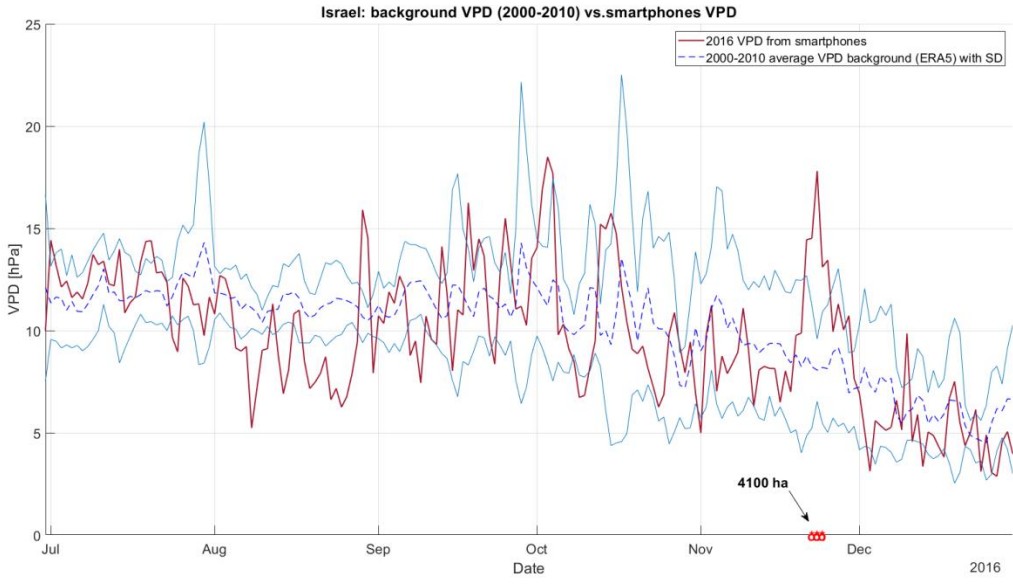

**Figure 10. Israel July-December 2016 daily mean calibrated smartphone VPD (red) vs. 10 years (2000-2010) daily mean**
**VPD for the same area from ERA5, with one standard deviation (light blue envelope) used as the background reference climatology of VPD. Fires indicated with red marks had a total burnt area 4100 ha.**

### 3.2 Portugal

According to "Forest Fires in Europe, Middle East and North Africa 2013-2016" reports (Schmuck et al., 2014; 2015; San-Miguel-Ayanz et al., 2016; 2017), in 2013 Portugal was severely affected by fires. More than 350 fires bigger than 40 ha occurred in Portugal, while most of the damage between July and September. The fire season in Portugal was more severe in 2013 due to the easterly flow over the Portuguese mainland providing hot continental air over the fire areas. The year 2016 is ranked second after 2013 with similar burned area but less fires in numbers (Table 2). The smartphone analysis for the Iberian Peninsula is shown in Figures 11-13 and is evaluated for latitudes 37-43 N and longitudes 9– 6W. Fires in this region were examined using data from the European Forest Fire Information System – EFFIS (https://effis.jrc.ec.europa.eu).

| Year | 2013 | 2014 | 2015 | 2016 |
|---|---|---|---|---|
| **Number of fires Portugal** | 19,291 | 7067 | 15,851 | 13,261 |
| **Total burnt area (ha)** | 152,756 | 19,929 | 64,443 | 161,522 |

**Table 2. Number of fires and total burnt area (ha) in Portugal for years 2013-2016.**

Figure 11 shows the temporal and spatial analysis, where the maps show the absolute VPD values calculated from smartphones for days in July 2013, starting with 2 July and with a 3-day interval between each map, while on 8 July a large wildfire occurred (>15,000 ha). Figure 12 shows the ΔVPD index, calculated using the monthly mean VPD of the discussed month (here July 2014, 2015 and 2016) and subtracting July 2013 **daily means** from the July 3– year **monthly mean**. Both Figure 11 and 12 show large extreme anomalies in VPD (> 8 hPa) in the day before and during the wildfire (8 July 2013). The regions in white did not have sufficient smartphone data for this analysis.

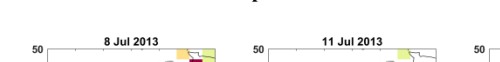

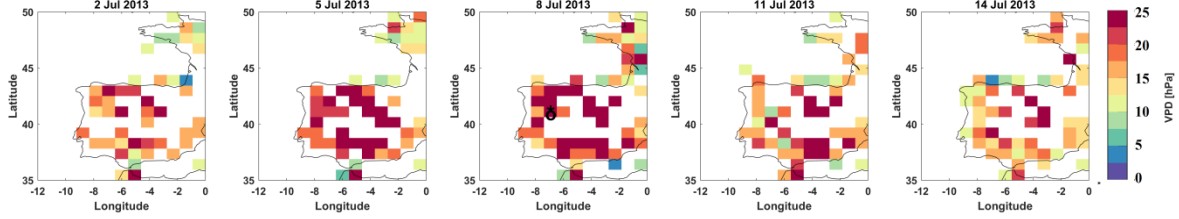

**Figure 11. Absolute VPD calculated from smartphones for days starting with 2 July and with a 3-day interval between each map. The wildfires occurred on 8 July 2013 with total burnt area >15,000 ha (locations of fires are marked with black symbols).**

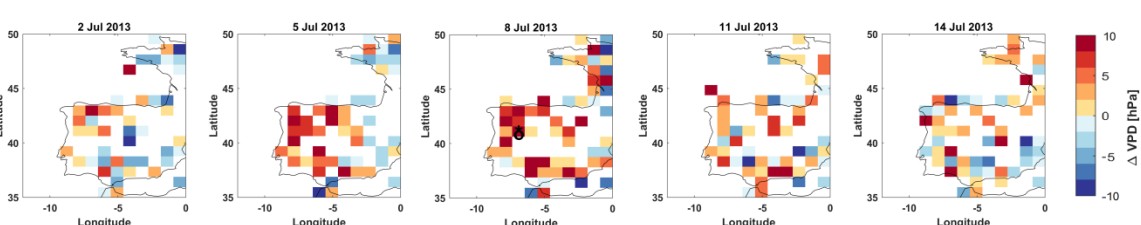


**Figure 12. VPD anomalies calculated from smartphones for days starting with 2 July and with a 3-day interval between each map. The fires occurred on 8 July 2013 with total burnt area >15,000 ha (locations of fires are marked with the black symbol).**

Figure 13 shows June to December 2013 daily VPD calculated from smartphones (red line), and the 2000-2010

daily mean VPD for the same area from ERA5 (blue dotted line), together with one standard deviation (blue lines). A significant increase in VPD is detected 2 weeks before the fire, when VPD stays high above the SD background envelope, and reaches a maximum anomaly 3 days before the large fire (5 July 2013), with VPD values of more than $6\sigma$ and staying around these values for 2 days after the fire started (10 July 2013). Such large anomalies are statistically significant above the 99.999% level, and suggest that dry and hot weather was surrounding the Portugal

region, drying fuels and increasing the fire risk.  Another interesting observation is the high VPD value around the end of November and the beginning of December 2013. December was surprisingly dry, with more than 510 fires reported and more than 1660 ha burnt around Portugal.

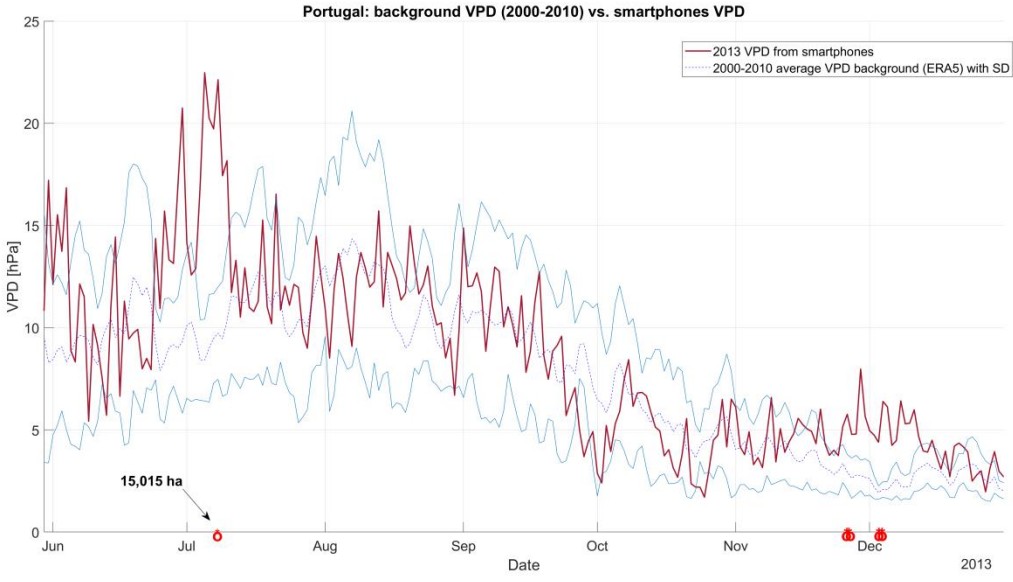

**Figure 13. Portugal June to December 2013 daily mean calibrated smartphone VPD (red) compared with 10 years (2000-**
**2010) daily mean VPD for the same area from ERA5 with one standard deviation (light blue envelope) used as the background reference climatology of VPD. Large fires are indicated with red symbols. In July 2013 the total burnt area >15,000 ha.  In late November and early December 2013 the total burnt area > 1660 ha.**

### 4. Conclusions and Discussion

We have analysed vapor pressure deficit (VPD) anomalies from crowd-sourced smartphones to show the potential of smartphone data for detecting significant drying anomalies related to wildfire events in two different locations in the Mediterranean region. The November 2016 fires in Israel together with the 2013 fires in Portugal were caused by dry weather that caused rapid drying out of surface fuel and local vegetation. The smartphone-calculated VPD anomalies in these two locations match closely the periods of severe dry weather and severe wildfires.

Large anomalies in VPD, calculated with calibrated smartphones T and RH, often occur days before the fires start, while continuing to rise during the fire periods, reaching 6 - 8 standard deviations above the mean, a highly statistically significant anomaly (>99.9999% significance). In our two case studies the VPD anomalies at their peaks were above +8 hPa. In this study we considered the daily mean VPD. This may introduce biases in the actual extreme VPD values, since some of the data may have been collected both outdoors (daytime) and indoors (nighttime). In addition, there have been recent studies highlighting the importance of nighttime VPD for the effectiveness of fire suppression operations (Balch et al., 2022). During nighttime, fuels may recover part of their moisture content; when VPD remains high at night, this process is hindered and fuels' flammability remains high, thus supporting the rapid spread of fire.

Not every significant increase in VPD leads to a fire, since without an ignition source a fire will not occur.. Furthermore, smartphones may give inaccurate readings since they are non-stationary, moving from desk to pocket, from indoor A/C to outdoors, sometimes from country side to urban heat islands. Of course, fire weather and warning depend on additional more complex parameters, such as solar radiation, types of fuel at the surface and subsurface, topography, cloud cover, ignition sources. However, many of these parameters may be reflected in the local temperature and humidity fields.

Our vision is to one day use crowd-sourced smartphone data to extract useful information that will help provide early warnings of fire hazards and risks. Such warnings could be supplied in real time to the public, to firefighters and emergency management authorities, at high spatial resolution and close to real-time. In the future, such warnings could be supplied on a smartphone application made available to users that contribute their data to the crowd-sourced early warning algorithm. As with mapping applications for traffic and hiking, future applications can be developed to supply smartphone sensor data to a central processing unit, and within a few seconds receive in return the fire hazard index to the users phone.

We propose that crowd-sourced smartphone data may eventually provide superior spatial resolution than regular meteorological networks as the density of smartphones over the world grows. Obviously, these data need to be first obtained in order to use them, but the coverage of this high-volume-high-density data is far greater than the coverage of stationary weather stations. This is especially true in developing countries where other sources of public early warnings are less available. In conclusion, we encourage the future use of smartphone data collected by the public in

helping to monitor extreme fire hazard conditions at high temporal and spatial resolution.  However, it should also be noted that this research was done using smartphone in the market between 2013-2017. The more recent smartphones have less sensors, and the relative humidity sensor is less common in recent smartphones in order to fulfil customers needs to make smartphones more robust against water.  In addition, due to the amount of data available to us from WeatherSignal, we averaged the data on 1x1 degree gridboxes.  While we have shown that when comparing with individual weather stations the agreement is significant (Figures 2 and 3), we recommend future analysis with better data should be performed at 0.1 degree spatial resolution, and 1 hour temporal resolution.

It is true that other observations networks also can supply the data supplied by the smartphone users, often at better quality.  However, the novelty of our methodology is using sensors that are commonly found in smartphones carried by the public, without specifically purchasing these sensors by the users, and often the users do not even know they are collecting these data.  The crowd-sourcing method presented here is not intended to replace conventional methods of collecting meteorological data, but as an additional tool to use, particularly since these sensors are non-stationary, with high density in populated areas, and particularly beneficial in developing countries where smartphones are quite well distributed related to meteorological fixed sensors.  Furthermore, the connectivity of smartphones via the cell phone networks is built in and transparent compared regular stationary observations networks.

Finally, to address privacy issues when using smartphone data, we suggest data collection Apps to save and supply only area-averaged values (superobs)(McNicholas and Mass, 2021) to researchers and users.  Supplying gridded data at one kilometre spatial resolution, and one hour temporal resolution would eliminate any privacy issues and user identification.

**Author contribution**: HS and CP designed the experiments and HS carried them out. HS prepared the manuscript with contributions from all co-authors. DRE and CM help with the editing and funding of this research.

**Acknowledgments**:  We would like to thank OpenSignal for supplying us the WeatherSignal data free of charge for this research.  In addition, we acknowledge the financial support for this research from the US-Israel Binational Science Foundation grant no. 2018186. The smartphone data can be obtained directly fro OpenSignal (https://www.opensignal.com/) via their Weather Signal App (https://weathersignal.en.uptodown.com/android).  The ERA5 data are available from the ECMWF (https://cds.climate.copernicus.eu/cdsapp#!/dataset/reanalysis-era5-single-levels?tab=overview).

**Competing interests**: The authors declare that they have no conflict of interest.

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
