# Peer review of "On the potential of using smartphone sensors for wildfire hazard estimation through Citizen Science"

_Natural Hazards and Earth System Sciences, 2023_

## Referee Comment (RC1)

Review of the manuscript nhess-2023-211

**On the potential of using smartphone sensors for wildfire hazard estimation**

By Hofit Shachaf and co-workers.

**Summary**: This manuscript reports about the results of a study where smartphone data collected from the Weather Signal App are used to assess the humidity and temperature state of the near surface atmosphere. These results are then related to the potential to follow and forecast fire weather. This is illustrated for cases in Israel, Portugal and Spain. The use of citizens science and crowdsourcing that uses smartphone data records is evolving. Earlier its focus was very much on temperature, but this manuscript brings the field a step further by developing a methodology to also use humidity observations. Therefore I think the work is suitable for publication after some revisions.

**Recommendation**: Revisions needed

**Major remarks**:

1. Ln 174: from here I get a little lost as reader. The manuscript is so far clear in the message that the goal of the paper is to use smartphone data to assess temperature and humidity to estimate wildfire risk. But from here it seems the reader is presented with lots of regressions between smartphone data between data from Spain, from Israel, from ERA5, from Portugal. Fitting local smartphone data records against ERA5 is risky since ERA5 is a rather coarse product (25 km), at least for this exercise. Hence landuse fractions are mixed, while both countries are at the coast. In addition, orography differences between the ERA5 grid cell and the local smartphone position can result in pressure and temperature differences. And Spain and Israel do have mountains. So why overall would one make so many correction formula's? If the goal is to show a correlation between smartphone data readings and professional products, then this can also be shown without fitting correction functions. Now it is very repetitive, while it is diluting the main message about the novelty of the method.

2. Discussion section: In my view the discussion section can be strengthened. Currently the authors are mainly bringing a very positive message about the potential for their method, but it is lacking critical notes that need to be ventilated, i.e.
   a. It remains unclear why only data from the Samsung S4 has been used, and whether the method is also applicable to other smartphone models. The S4 model is very much outdated and hardly used anymore. The paper should discuss this deficiency. As far as I know smartphone manufacturers are on a track to reduce the number of sensors in their smartphone models, in order to fulfil customers needs to make the smartphones more robust against water.
   b. The paper should be more critical on the aspect of using nearly live smartphone weather data to follow and nowcast fire risk. Personally I think is rather far fetched, and an ideal picture. As far as I know making weather data available for scientists involves quite some handwork for OpenSignal, so a live stream would not be possible in practise. Please discuss.
   c. Related to point b, could the authors comment on the added value of their method compared to existing platforms like Weatherunderground.com, NetAtmo.com, and https://wow.metoffice.gov.uk/ that distribute citizen science data from relatively dense networks, more or less live or semi-live. Are these platforms already offering all information needed for fire risk assessment?
   d. The paper does not reflect so much that the dynamics of wild fires is much more complex than just indices that are presented. Pyrocumulus and pyrocumulonimbus clouds trigger very much their own meteorological conditions through feedbacks in cloud cover, solar radiation, up- and downdrafts, local winds. Moreover the wild fire risks also depend on the fuel amount and type on the surface and the sub-surface. Lots of recent knowledge has been developed about these dynamics. It would be good if the manuscript could reflect some of the complexity involved, and especially how these complexities would prohibit the use of the smartphone data in an meaningful way, or the other way around -i.e. whether the authors think there is also potential in the methods to follow individual Pyrocumulonimbus clouds.

Minor remarks on the next page

**Minor remarks**:

Ln 25: smartphone ownership growth is certainly not exponential anymore. See e.g. this graph (from https://explodingtopics.com/blog/smartphone-stats). Please reword.

[Figure]

Ln 52: Note temperature estimates were also made for contrasting urban form for Sao Paolo in Droste et al (2017; https://journals.ametsoc.org/view/journals/atot/34/9/jtech-d-16-0150.1.xml).

Ln 79: Please put the table caption above the table.

Ln 100: perhaps add a word or two to clarify that the IMS stations are the professional or reference stations.

Ln 123: The authors report here the sensitivity and accuracy of the sensors as provided by the manufacturer. In the recent study by Noyola et al. (https://www.frontiersin.org/articles/10.3389/fenvs.2021.673937/full ), the authors have used the Samsung S4 in the field too, and have performed experiments to determine the response time of both the temperature and RH sensors. Likely there is useful information in that paper for the current study as well. In a practical setup the RH error appeared to be higher than reported in the current manuscript. The cold bias in Fig 2a is about similar to the one reported in Noyola et al.

Ln 164: The calibration was done also using a simple linear regression model. Could authors explain whether a linear regression on a conserved variable like specific humidity would perhaps have worked better? The relation between RH and temperature is highly non-linear (Clausius-Clapeyron), so perhaps a linear correction of RH might result in a biased outcome. Please comment.

Ln 170, Fig 3: please label the panels a and b and adjust the figure caption. In the panel on the right the y axis can start at RH =30% to show a better contrast between the two lines.

Fig 4: for the linear regression formula that is shown, please add in the caption that the temperature input should be in degrees C, not Kelvin.

Fig 4b,d: in panel b the $R^2$ is in 2 decimals, while in panel d it is 1 decimal. Please make consistent, to respect the significance of the results.

Ln 217: note that the ECMWF model also has obvious biases in wintertime, often reporting warm biases too, so be careful with using ECMWF or ERA5 as "the truth" for these winter conditions and stable boundary layers.

Ln 240: the same RH at different temperatures results in very different VPD values. Please reword, since RH is a derived variable from conserved variables like water vapor pressure (at the surface) or specific humidity and temperature. Hence the temperature and the vapor pressure themselves govern what is the RH, not the other way around.

Ln 265: IMS has already been introduced before.

Ln 273: November 2013, 2015 and 2015. 2015 twice?

Ln 353: …We are now attempting to monitor wind speed using smartphone horizontal pressure gradient data between different locations,… Please remove, this is somewhat speculative and I find this beyond the scope of the current paper. In addition wild fires trigger their own wind field/circulations so the wind field will not always follow the large scale field triggered by synoptic pressure gradients.

Acknowledgements: I miss a data availability statement.

---

## Author Comment (AC1)

Review of the manuscript nhess-2023-211

**On the potential of using smartphone sensors for wildfire hazard estimation**
By Hofit Shachaf and co-workers.

**Summary**: This manuscript reports about the results of a study where smartphone data collected from the Weather Signal App are used to assess the humidity and temperature state of the near surface atmosphere. These results are then related to the potential to follow and forecast fire weather. This is illustrated for cases in Israel, Portugal and Spain. The use of citizens science and crowdsourcing that uses smartphone data records is evolving. Earlier its focus was very much on temperature, but this manuscript brings the field a step further by developing a methodology to also use humidity observations. Therefore I think the work is suitable for publication after some revisions.

We thank the Reviewer for his/her comments, and the positive criticism was taken into consideration during the revision of the manuscript. The detailed response is provided below.

**Recommendation**: Revisions needed

**Major remarks:**

1. Ln 174: from here I get a little lost as reader. The manuscript is so far clear in the message that the goal of the paper is to use smartphone data to assess temperature and humidity to estimate wildfire risk. But from here it seems the reader is presented with lots of regressions between smartphone data between data from Spain, from Israel, from ERA5, from Portugal. Fitting local smartphone data records against ERA5 is risky since ERA5 is a rather coarse product (25 km), at least for this exercise. Hence landuse fractions are mixed, while both countries are at the coast. In addition, orography differences between the ERA5 grid cell and the local smartphone position can result in pressure and temperature differences. And Spain and Israel do have mountains. So why overall would one make so many correction formula's? If the goal is to show a correlation between smartphone data readings and professional products, then this can also be shown without fitting correction functions. Now it is very repetitive, while it is diluting the main message about the novelty of the method.

These regressions using the ERA5 data are simply to strengthen the argument that smartphones **can** supply useful information (both temperature and RH) for meteorological research. The regression functions were used for "training" and "correcting" the smartphone data, and later used in the "testing" mode for the fire events analyzed. If the reviewer thinks this additional analysis with the ERA5 is not necessary, we can remove it from the paper.

2. Discussion section: In my view the discussion section can be strengthened. Currently the authors are mainly bringing a very positive message about the potential for their method, but it is lacking critical notes that need to be ventilated, i.e.

   a. It remains unclear why only data from the Samsung S4 has been used, and whether the method is also applicable to other smartphone models. The S4 model is very much outdated and hardly used anymore. The paper should discuss this deficiency. As far as I know smartphone manufacturers are on a track to reduce the number of sensors in their smartphone models, in order to fulfil customers needs to make the smartphones more robust against water.

   Good point. However, the Samsung Galaxy was used only for our control tests. The Weather Signal data is obtained from ALL available Android smartphones used during 2013-2017. We have added some discussion on this point at the end of the paper.

b. The paper should be more critical on the aspect of using nearly live smartphone weather data to follow and nowcast fire risk. Personally I think is rather far fetched, and an ideal picture. As far as I know making weather data available for scientists involves quite some handwork for OpenSignal, so a live stream would not be possible in practise. Please discuss.

We think this is less of a problem. We have developed our own smartphone App that can transfer data in real time to our central operations computer, where the analysis of VPD takes additional seconds. So just as Google Maps can track the location of your position in near real time, and you can talk on your smartphone while moving, the smartphone sensors can also send their data in real time for analysis and display on the users smartphone. We have added some discussion on this point.

c. Related to point b, could the authors comment on the added value of their method compared to existing platforms like Weatherunderground.com, NetAtmo.com, and https://wow.metoffice.gov.uk/ that distribute citizen science data from relatively dense networks, more or less live or semi-live. Are these platforms already offering all information needed for fire risk assessment?

All existing networks can provide the data needed to estimate fire risks according the VPD calculations in our paper. The novelty in our study is using sensors that are commonly found in smartphones carried by the public, without specifically purchasing these sensors, and often with the users not even knowing they are collecting these data. The crowd-sourcing method presented here is not intended to replace conventional methods of collecting meteorological data, but as an additional tool to use, particularly since these sensors are non-stationary, with high density in populated areas, and particularly beneficial in developing countries where smartphones are quite well distributed relative to official fixed meteorological sensors. Furthermore, the connectivity of smartphones via the cell phone networks is built in and transparent compared to regular stationary observations networks. We have added this to the discussion section.

d. The paper does not reflect so much that the dynamics of wild fires is much more complex than just indices that are presented. Pyrocumulus and pyrocumulonimbus clouds trigger very much their own meteorological conditions through feedbacks in cloud cover, solar radiation, up- and downdrafts, local winds. Moreover the wild fire risks also depend on the fuel amount and type on the surface and the sub-surface. Lots of recent knowledge has been developed about these dynamics. It would be good if the manuscript could reflect some of the complexity involved, and especially how these complexities would prohibit the use of the smartphone data in an meaningful way, or the other way around - i.e. whether the authors think there is also potential in the methods to follow individual Pyrocumulonimbus clouds.

Obviously, smartphone data is limited to what is observed by the sensors in the phones. We are working on a new idea related to wind speed based on horizontal pressure gradients using numerous smartphone locations and data. But other more complex indices cannot be obtained using only smartphone data. We have added some additional comments about the complexity of the issue in the discussion.

Regarding pyro cumulus clouds and possible additional ignitions from the fire plumes, this is a result of the fires **after** they already occur. This is beyond the scope of this paper.

**Minor remarks:**
Ln 25: smartphone ownership growth is certainly not exponential anymore. See e.g. this graph (from https://explodingtopics.com/blog/smartphone-stats). Please reword.

We have changed "exponential" to "rapid."

[Figure]

Ln 52: Note temperature estimates were also made for contrasting urban form for Sao Paolo in Droste et al (2017; https://journals.ametsoc.org/view/journals/atot/34/9/jtech-d-16-0150.1.xml). Reference added to revised manuscript.

Ln 79: Please put the table caption above the table.
Done

Ln 100: perhaps add a word or two to clarify that the IMS stations are the professional or reference stations.
Added to the caption of Fig. 1

Ln 123: The authors report here the sensitivity and accuracy of the sensors as provided by the manufacturer. In the recent study by Noyola et al. (https://www.frontiersin.org/articles/10.3389/fenvs.2021.673937/full), the authors have used the Samsung S4 in the field too, and have performed experiments to determine the response time of both the temperature and RH sensors. Likely there is useful information in that paper for the current study as well. In a practical setup the RH error appeared to be higher than reported in the current manuscript. The cold bias in Fig 2a is about similar to the one reported in Noyola et al.
Thank you for this reference, however it is from Cabrera et al., 2021. We have added the reference to the revised manuscript.

Ln 164: The calibration was done also using a simple linear regression model. Could authors explain whether a linear regression on a conserved variable like specific humidity would perhaps have worked better? The relation between RH and temperature is highly non-linear (Clausius-Clapeyron), so perhaps a linear correction of RH might result in a biased outcome. Please comment.

Unfortunately, we only have RH and not specific humidity from the smartphones. For fires RH is more important than SH due to the VPD that results in drying of the vegetation. However, for atmospheric temperatures in the summer (fire season) the CC relationship can be well approximated by a linear function between SH and T. Clausius-Clapeyron does not show the relationship between RH and T.

Ln 170, Fig 3: please label the panels a and b and adjust the figure caption. In the panel on the right the y axis can start at RH =30% to show a better contrast between the two lines.
Done

Fig 4: for the linear regression formula that is shown, please add in the caption that the temperature input should be in degrees C, not Kelvin.
Done

Fig 4b,d: in panel b the $R^2$ is in 2 decimals, while in panel d it is 1 decimal. Please make consistent, to respect the significance of the results.
Done

Ln 217: note that the ECWMF model also has obvious biases in wintertime, often reporting warm biases too, so be careful with using ECMWF or ERA5 as "the truth" for these winter conditions and stable boundary layers.
Noted, thanks.

Ln 240: the same RH at different temperatures results in very different VPD values. Please reword, since RH is a derived variable from conserved variables like water vapor pressure (at the surface) or specific humidity and temperature. Hence the temperature and the vapor pressure themselves govern what is the RH, not the other way around.
Corrected.

Ln 265: IMS has already been introduced before.
Corrected

Ln 273: November 2013, 2015 and 2015. 2015 twice?
Corrected

Ln 353: …We are now attempting to monitor wind speed using smartphone horizontal pressure gradient data between different locations,… Please remove, this is somewhat speculative and I find this beyond the scope of the current paper. In addition wild fires trigger their own wind field/circulations so the wind field will not always follow the large scale field triggered by synoptic pressure gradients.
Done

Acknowledgements: I miss a data availability statement.

added

---

## Author Comment (AC2)

In this article, the authors present a useful tool available to virtually anyone (Android cell phones) to address the challenge that poses wildfires with 2 study cases in Portugal and Israel. Overall, it is well written in academic English, easy to follow, provides a large range of well-done figures, and gives 2 interesting study cases with thorough research.

We thank this reviewer for the positive response. We have addressed the comments below.

My comments to improve the paper are more focused on the main theme of the article because the technical issues were treated by reviewer 1 and resolved by the authors:

1.      I think it could be necessary to work on the title, abstract, and discussion about what could be the most interesting part of the experiment which is the citizen science. It could bring novelty to the topic of wildfire hazard monitoring empowering citizens of all ages. A possible title could be: "Experiencing the Potential of Use of Android Smartphone Sensors in Wildfire Hazard Estimation through Citizen Science".

This is an interesting idea emphasizing citizen science. We have now changed the title to: "On the potential of using smartphone sensors for wildfire hazard estimation through Citizen Science."

2. Also, in my opinion, it was not very clear in the introduction, methodology, and discussion which could be the main reader of this article. Paragraphs 355-370 enumerated some potential readers, but they are mentioned at the end of the article and did not permeate through the entire manuscript. Was it written for citizen science advocates? For wildfire researchers? For public policy specialists in wildfires, climate, and disasters? For engineers who work in big data? For governments in low-income countries who cannot afford an abundance of meteorological stations? This lack of definition debilitated an otherwise good article and must be clearly stated throughout the manuscript, please consider a major revision from peers in those abovementioned topics.

While this paper is a scientific paper submitted to a scientific journal, we agree that there may be many possible interested parties in the results of the study. Hence, we have added a sentence up front in the abstract related to the possible users of this technology.

3. It will be very important to add in paragraph 145 that the WeatherSignal App does not operate in IOS (iPhone) and it is available only for Android smartphones. This should be repeated in the limitations of this study (paragraphs 365 and 370).

We have added a sentence to note this.

3. In aspects of formatting, please use the table model from the journal.

Done

---

## Author Comment (AC3)

REVIEW OF MANUSCRIPT NHESS-2023-211

"On the potential of using smartphone sensors for wildfire hazard estimation"

We thank the reviewer for his/her constructive criticism and comments on the paper. We have addressed all comments below and have made the appropriate changes to the revised manuscript.

**Summary and recommendation**

The manuscript presents a study of using smartphone-based T/RH measurements to estimate VPD and subsequently assess wildfire hazard. The use of smartphone-based data is certainly a novel aspect of the work presented and something that the wildfire community could examine in the future for assessing wildfire hazard or/and fire danger. However, despite the arguments presented by the authors in the manuscript, I, unfortunately, remain skeptical towards the added value of the proposed approach. I recommend to reconsider the manuscript for publication subject to a major revision that, in my view, would allow for better highlighting any possible added value of the proposed use of smartphone-based data for assessing the conduciveness of weather to wildfires.

This paper presents a proof-of-concept idea that is novel and should be expanded on by future researchers when better smartphone data is available. At the moment we are limited by the accessibility to smartphone data due to privacy issues, and hence this paper used anonymous data provided by a third party. With all the limitations of the data, we still feel this paper is worthy of publication due to the novelty of using smartphone sensors in estimating fire hazard and danger. The added value will occur when the high spatial resolution data will be available in real time to fire weather experts and forecast centers. This may be available to companies like Google, IBM, Microsoft, and others. So this paper provides a new direction for monitoring in real time fire weather indicators.

**Major remarks**

1. In L41, the authors claim that smartphone microsensor data "*ay provide additional and highly complimentary data*", as compared to traditional weather station data. Further, in L91-93, the authors state that "*smartphone sensors have the potential for providing useful information about VPD at high spatial resolution and high temporal resolution even in remote areas with few official weather stations*".

I am very worried that the analysis conducted by the authors and the results presented do not support the above two arguments made to support the added value of using smartphone data.

First, the authors conduct their analysis on a 1 x 1 degree grid, on which they interpolate the smartphone-based VPD estimates. This spatial resolution is too coarse for any assessment of wildfire hazard/danger and hence, the presented results cannot be used for supporting the authors' claim about smartphone microsensors providing information about VPD at high spatial resolution.

We started our analysis in the paper with comparisons with single weather station data in Israel (control experiments Figure 2) and in Spain (see Figure 3). Due to the difficulty of obtaining surface observations at the specific locations of the smartphones, we performed the control experiments (Figure 2) to show the ability of the smartphone sensors to measure environmental parameters reliably, and then we used two meteorological station in Spain for additional comparisons (point data) in Figure 3, showing the added value of the smartphone data to the normal meteorological stations, after simple calibrations.

The additional 1x1 degree analysis was performed on the regional scale due to the lack of smartphone data in many locations, and the variability of the data in space and time. We do not do any interpolation, only averaging. If there is no data, we do not present a value for VPD. For regions with lots of data we could use 0.1 degree boxes (~10km), but for the regional plots (due to a small amount of smartphones) we used 1 degree grids for better visualisation. We were also limited in in our case studies since the smartphone data available to us was only for the period of 2013-2017. Again, this paper is to show the value of these smartphone data in detecting anomalies in VPD. Having millions of data points in the future would allow us to show the VPD on much finer spatial scales. This is the vision for the future, but not possible with the data we have now.

Second, the authors computed gridded VPD data by taking the daily mean values of T and RH. This is a rather crude approach that introduces significant implications. By considering the daily mean T/RH values, the authors put together data that may have been collected both outdoors (daytime) and indoors (nighttime). In addition, there have been recent studies highlighting the importance of nighttime VPD for the effectiveness of fire suppression operations. During nighttime, fuels may recover part of their moisture content; when VPD remains high at night, this process is hindered and fuels' flammability remains high, thus supporting the rapid spread of fire. By taking a daily mean T/RH value to estimate a kind of daily mean VPD, this important information is neglected.

Good point. We appreciate and understand the finer diurnal dynamics of VPD and the impact on the fire hazard. Unfortunately, as mentioned above our data was limited to the WeatherSignal App data, and hence separating the data into day and night would have resulted in even less data per region and gridbox. We agree that ideally, we should separate the data to even hourly if possible. We have added

some discussion on this aspect of the diurnal VPD in the revised paper. We have added the reference to the recent paper by Balch et al. (2022)

Balch, J.K., Abatzoglou, J.T., Joseph, M.B. *et al.* Warming weakens the night-time barrier to global fire. *Nature* **602**, 442–448 (2022). https://doi.org/10.1038/s41586-021-04325-1

Third, smartphone data availability is directly related to the population density. Where more people gather, more data will be available. This typically includes populated areas and very less often remote areas. In addition, when the smartphone data are gathered from people located in an urban area, one should take into account that conditions may be significantly different from a remote, mountainous area. By using these data to assess conditions in another totally different environment (in terms of land cover, topography, etc.), one should be aware of the significant uncertainty introduced through the extrapolation of the data.

This comment is not clear to us. No extrapolation is done regionally. The training and testing data are for the same regions. The VPD is calculated for each gridbox individually, and ideally the calibration of the smartphone data will be done for each gridbox separately. Hence, an urban area will have it's own calibration and calculation of VPD, while a remote rural region will have it's own calibration and VPD values. However, the primary use and benefit of such a novel technique would be in remote regions where we have less meteorological measurements and infrastructures (and where fires may be more important) while also in developing countries, where in some locations there are more people with mobile phones than people with electricity or running water. See infographics below.

[Figure]

In summary, I highly encourage the authors to reconsider the entire structure of their work to better highlighted the possible added value of smartphone data for wildfire applications. Some suggestions may include:

- Decrease the spatial resolution of the grid on to which the smartphone data are interpolated, possibly to match the resolution of one of the publicly available reanalysis datasets (e.g., ERA5 at 0.25 x 0.25 degrees).

  As mentioned above, we have shown the agreement between the smartphones and individual weather stations in our paper (Figures 2 and 3). However, when looking at regional smartphone data from WeatherSignal, the distribution is not uniform in space or time. We have tried to grid the data at 0.25 degrees, but due to the low number of data points in each 0.25 gridbox, the results are much noisier. Given new data sets that may be available to Google, IBM, and others, this may be possible in the future. We added some discussion on this issue in the revised manuscript.

- Comparison of the smartphone-based VPD against ERA5-based (or any other data source) VPD and VPD obtained from interpolation from weather stations. This would allow evaluating whether smartphones can provide additional information or not.

  In the paper we DO present the comparison of VPD from the smartphones and ERA5 in Figures 10 (Israel) and 13 (Portugal)

- Refrain from averaging daily T/RH values and focus on examining nighttime/daytime data.

  As mentioned above, the scarcity of WeatherSignal data in each location limits what we can do with the present data. We added some discussion on this point for future researchers in this field. We had our hands tied since we did not collect the data ourselves, but were using a data set supplied to us, and collected originally for other purposes.

---

## Referee Report (RR1)

Review of the revised version of nhess-2023-211

On the potential of using smartphone sensors for wildfire hazard estimation through Citizen Science

Summary: The authors have thoroughly responded to the concerns that I raised on the original version of the manuscript. I find the responses also valid and to the point. However, I find at this moment only the reviewers benefit from the reflections provided in the responses to the reviewer's remarks, but the readership does not see it. So I recommend to make another revision in which these reflections are entered into the main manuscript. Also I have some additional concerns about the quality of the figures.

Recommendation: revisions needed

Major remarks

1. I find the reflections and answers to my raised concerns as provided in the rebuttal very appropriate and valid. However, I find these reflections need to be transferred to the main manuscript. Indicating the strengths and weaknesses or open issues is not a weakness to the paper, but helps to put the results in context and also to formulate a research agenda for the community. Especially the answers to the concerns 2a-2d are also useful for the readership of the paper, so please include these reflections in the (discussion section of the) paper.

2. Quality of the figures. I had a special look at the quality of the figures. Since these are the working horses of the paper to convey the message you want to bring forward, they should be of high quality. Here are my concerns:
   -Fig 2: Top header can be removed and its information can be transferred to the figure caption. Lines should be thicker and the font size along the axes larger.
   -Fig 3: please label the two panels a and b, adjust the caption accordingly and make sure both panels are referred to in the text. The y axis of panel b (on the right) can start at RH=30% to enhance the difference between the two lines. Why does this plot have a grid, while the panels in Fig 2 do not have it. Please make consistent. Top header can be removed and its information can be transferred to the figure caption.
   -Fig 4: Top header can be removed and its information can be transferred to the figure caption. Lines should be thicker and the font size along the axes larger.
   -Fig 5: Top header can be removed and its information can be transferred to the figure caption. Lines should be thicker and the font size along the axes larger.
   -Fig 6: Top header can be removed and its information can be transferred to the figure caption. Lines should be thicker and the font size along the axes larger.
   -Fig 7: Top header can be removed and its information can be transferred to the figure caption. Lines should be thicker and the font size along the axes larger. Y axis can start at RH=30% in both panels.
   -Fig 8+9: label the panels a-e. The font size on the axes need to be enlarged.

-Fig8+9: the paper has quite many figures. Can figure 8 and 9 be combined into one figure? The same holds for Figs 11+12.
-Fig 10 and 13: Top header can be removed and its information can be transferred to the figure caption. Lines should be much thicker and the font size along the axes larger. Figure does not seem to be sharp here.

---

## Author Response (AR2)

**Review of the revised version of nhess-2023-211**

On the potential of using smartphone sensors for wildfire hazard estimation through Citizen Science

Summary: The authors have thoroughly responded to the concerns that I raised on the original version of the manuscript. I find the responses also valid and to the point. However, I find at this moment only the reviewers benefit from the reflections provided in the responses to the reviewer's remarks, but the readership does not see it. So I recommend to make another revision in which these reflections are entered into the main manuscript. Also I have some additional concerns about the quality of the figures.

Thank you for this important comment. We have added the discussion to the revised manuscript.

Recommendation: revisions needed

Major remarks

1. I find the reflections and answers to my raised concerns as provided in the rebuttal very appropriate and valid. However, I find these reflections need to be transferred to the main manuscript. Indicating the strengths and weaknesses or open issues is not a weakness to the paper, but helps to put the results in context and also to formulate a research agenda for the community. Especially the answers to the concerns 2a-2d are also useful for the readership of the paper, so please include these reflections in the (discussion section of the) paper.

Discussion added to the revised paper.

2. Quality of the figures. I had a special look at the quality of the figures. Since these are the working horses of the paper to convey the message you want to bring forward, they should be of high quality. Here are my concerns:

-Fig 2: Top header can be removed and its information can be transferred to the figure caption. Lines should be thicker and the font size along the axes larger.

Figure revised and improved.

-Fig 3: please label the two panels a and b, adjust the caption accordingly and make sure both panels are referred to in the text. The y axis of panel b (on the right) can start at RH=30% to enhance the difference between the two lines. Why does this plot have a grid, while the panels in Fig 2 do not have it. Please make consistent. Top header can be removed and its information can be transferred to the figure caption.

Changes made to figure and caption.

-Fig 4: Top header can be removed and its information can be transferred to the figure caption. Lines should be thicker and the font size along the axes larger.

Changes made to figure and caption.

-Fig 5: Top header can be removed and its information can be transferred to the figure caption. Lines should be thicker and the font size along the axes larger.

Changes made to figure and caption.

-Fig 6: Top header can be removed and its information can be transferred to the figure caption. Lines should be thicker and the font size along the axes larger.

As above

-Fig 7: Top header can be removed and its information can be transferred to the figure caption. Lines should be thicker and the font size along the axes larger. Y axis can start at RH=30% in both panels.

Changes made

-Fig 8+9: label the panels a-e. The font size on the axes need to be enlarged.

Done

Fig8+9: the paper has quite many figures. Can figure 8 and 9 be combined into one figure? The same holds for Figs 11+12.

Yes, we have combined the figures.

Fig 10 and 13: Top header can be removed and its information can be transferred to the figure caption. Lines should be much thicker and the font size along the axes larger. Figure does not seem to be sharp here.

Figures have been improved.